# Use of Olive Industry By-Products for Value-Added Food Development

María E. García-Pastor [ID], Marina Ródenas-Soriano, Alicia Dobón-Suárez, Pedro J. Zapata [ID] and María J. Giménez *[ID]

Department of Food Technology, EPSO, University Miguel Hernández, Ctra. Beniel km. 3.2, 03312 Alicante, Spain
* Correspondence: maria.gimenezt@umh.es; Tel.: +34-966749798

**Abstract:** The olive industry involves environmental problems of special relevance, such as the generation of residual brines after the pickling process. Moreover, in the pitting and filling operations of the olives, fatty residues are generated that end up being washed away in the wastewater of these industries. The novelty of this work is based on the extraction of high value-added compounds from residues resulting from the industrialization process of olives, whose content in fatty acids, phenolic compounds and other substances could make them susceptible to being considered as a raw material of interest for the development or enrichment of other foods. The results obtained determined that the physico-chemical and functional characteristics of the oil and the brine, generated as by-products in the olive industry, indicated their potential as raw materials. For this reason, new formulations of the filling of olives (anchovy, red pepper, and lemon flavour) were developed, using the residual oil as a by-product, which showed greater global acceptability by consumers compared to a commercial stuffing made with sodium alginate. In addition, wastewater was used as a brine by-product to pickle three types of vegetables: carrot, cauliflower, and onion. All pickled vegetables showed the highest phenolic content and a higher overall liking, the differences being significant with respect to commercial reference. In conclusion, the results obtained could allow for the conversion of a by-product into a co-product, partially solving an environmental problem, and providing added value to the final product.

**Keywords:** antioxidants; brine; emulsion; oil; pickles

## 1. Introduction

Olive production is a relevant activity in areas with Mediterranean-type climates, where the availability of irrigation water or rainfall is limited. The production of olives is located mainly in countries of the Mediterranean basin, that corresponds to more than 90% of the surface of olive production in the world. Spain is the world leader in the production and export of table olives, with 62% of EU production and 17% of worldwide olive production. Specifically, Extremadura produces around 13% of the total and Andalusia produces close to 80%, with Seville as the main producing province that comprises approximately 58% of the total national production [1].

Table olives are the most widely consumed fermented food in Mediterranean countries [2]. The table olive industry generates large amounts of wastewater due to the alkaline treatment, fermentation and washing steps. Moreover, in the olive pitting and stuffing operations, fatty residues are generated that end up being disposed in the wastewater. Therefore, these wastewaters are characterized by a high content of organic matter, a high percentage of suspended solids and fats, an acidic or alkaline pH, high conductivity due to its high salt content, and colored waters due to the presence of polyphenolics. The presence of these compounds in wastewaters make their treatment difficult, leading to a relevant environmental problem with a complicated technological, economic, and social solution [3,4].

However, the high amounts of polyphenols (0.5–24 g L$^{-1}$) could show potential health benefits [5] related to their scavenger properties against free radicals and reactive oxygen forms (ROS); their ability to act as chelators of heavy metals (especially iron); and their ability to inhibit lipoxygenase. The main phenolic compounds present in olive mill wastewaters are hydroxytyrosol, tyrosol, caffeic acid, *p*-coumaric acid, vanillic acid, syringic acid, gallic acid, luteolin, quercetin, cyanidin, verbascoside and some polymeric compounds [6]. Research has been carried out to obtain phenolic compounds, since they show numerous therapeutic capacities of great interest, such as anti-inflammatory, antioxidant, antiaging, cardioprotective, and anticancer properties [7,8]. Several studies have been carried out about the extraction of phenolic compounds through membrane technologies for their application in the food, cosmetic and/or pharmaceutical industry [9–12]. As far as we know, olive oil and brine by-products have not been used to develop new products at the industrial level. On the other hand, these wastewaters also have fatty acids such as oleic, palmitic, and linoleic acids as main compounds, as identified by Padalino et al. [13]. The growing popularity of olive oil wastes is also attributed to the content of oleic acid, which is considered a valuable source of antioxidants in the human diet. During its production, certain amounts of fatty acids remain in the olive oil by-products and waste [14,15]. Therefore, the olive mill wastewaters may be a suitable source of valuable compounds that could be used to transform an agro-industrial wastewater into useful and relevant ingredients [16].

In recent years, the concept of the circular economy and waste valorization has gained relevance to reduce waste and conserve resources, making more efficient utilization and revalorization of products to obtain value-added products, contributing to a more sustainable economy, and reducing environmental problems [17]. In this sense, recent investigations have focused on the search for new opportunities for the use of these residues, such as agricultural applications and source of bioactive substances [18], functional food ingredient [19], and use as a source of biomass [20]. In recent studies, olive oil by-products have been added as concentrates or ingredients in the formulation of novel foods in different agro-food supply chains. Galanakis et al. [21] collected data related to the addition of oil mill extracts (but also of other oil industrial by-products) to fortify meat and meat products. In addition, several studies have tested phenol extracts in dairy products to enhance antioxidant activity and stabilize the food products [22–25]. On the other hand, other authors [21,26,27] also enriched bakery products by adding phenolic extracts from olive mill wastewater. Finally, a potential use of olive mill wastewater has been reported in the preparation of functional beverages [28]. According to the circular economy production model, two by-products (oil and brine) from olive industry wastewater were included in the food chain as raw materials with high added value, involving the concept of sharing, leasing, reusing, repairing, renewing, and recycling existing materials and products for as long as possible. Thus, the main objective of this research work was the characterization, reuse and/or recovery of two by-products generated in the olive industry, oil (fat residue) and brine (aqueous residue), for the development of food products with added value: new formulations using oil by-products for the preparation of stuffed olives (anchovy, red pepper, and lemon) and saline concentrates from the by-product of the brine for vegetable pickles (Figure 1).

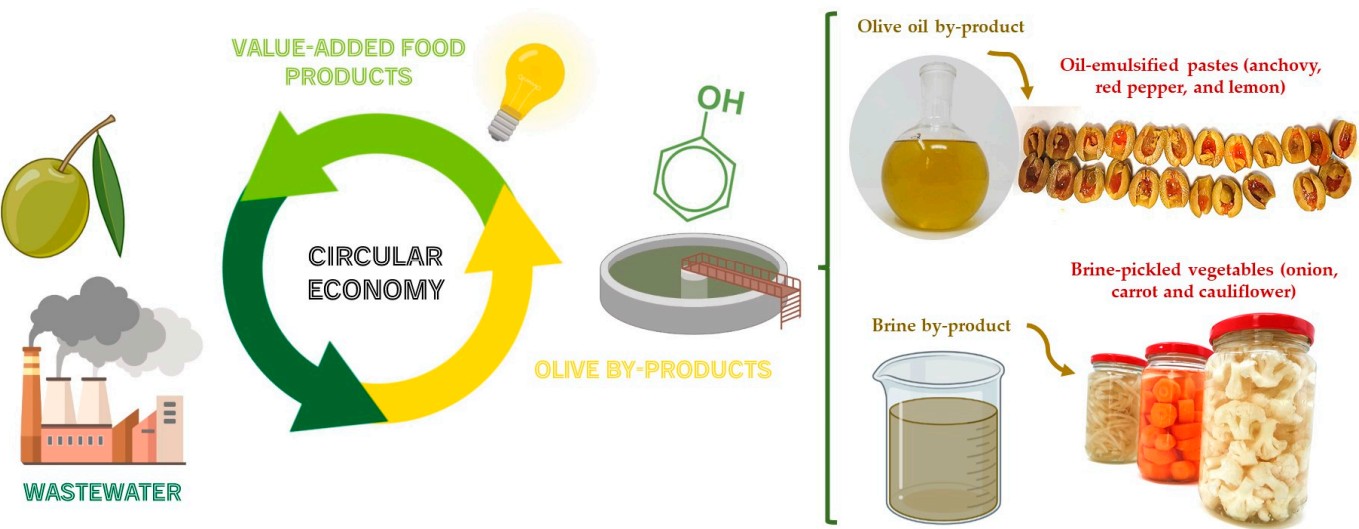

**Figure 1.** Summary of the general concept of circular economy and the main aim of the present study for the revaluation of olive industry by-products.

## 2. Materials and Methods

### 2.1. Experimental Design

2.1.1. Separation of Olive Industry By-Products

In order to reduce the contamination caused by by-products or wastewater (oil + brine) originating in the stuffed olive processing phases (pitting and stuffing), the by-products were subjected to a centrifugation process for the separation of both phases. The experiment was carried out with wastewater of the 'Hojiblanca' cultivar olive-pitting and stuffing line. For this, a mill centrifuge was used, in which the mixture of oil and water constantly recirculated, and both phases, the aqueous phase (brine) and fatty phase (oil), were separated, obtaining a yield close to 80% (Supplementary Figure S1). The wastewater obtained from the olive-pitting and filling process went through the centrifuge, where oil and brine by-products were extracted, and it was recirculated all week and discharged at the end of the week. This centrifuge had a flowrate of 800 L h$^{-1}$ and worked 7 h per day on average; therefore, to carry out each of the tests, it was estimated that 5600 L passed through the centrifuge per day and the oil recovered was calculated based on the liters of centrifuged wastewater. After all the tests were performed, the mill centrifuge separated up to 80% of the waste, that is, it did not extract all the oil that the wastewater contained. In this sense, the final efficiency recovery of oil in this process was around 1.6 and 2.2% from the total wastewater processed. This percentage may vary depending on the processing line and the cultivar of olives used. The physico-chemical and functional characterization of the by-products obtained was performed. The fatty acid profile, colour, viscosity and total phenolics were then analysed in the oil by-product. Moreover, pH, conductivity, water activity (aw) and total phenolic content analyses were carried out in the brine by-product.

2.1.2. Development of Value-Added Products

For the development of the new products, both by-products recovered previously were used. The fatty fraction obtained was used to elaborate new formulations of stuffing or paste for olives, while the aqueous fraction was used to develop different vegetable pickled products. On the one hand, the new stuffing emulsions were made by substituting the sodium alginate used by the companies (control formulation) with the oil obtained as a by-product, which acted as an emulsifier for the emulsion or stuffing paste. The formulations developed in this research work were made using anchovy, pepper, and lemon commercial pastes, as can be seen in Table 1. Other food additives used, such as citric acid, gelatin, sodium alginate or guar gum, were provided by Prosur S.L. (Murcia, Spain). The different

formulations prepared were made with distilled water (Supplementary Figure S2). All experimental tests were carried out in the laboratory at room temperature ($25 \pm 1\ °C$).

**Table 1.** Ingredients of the stuffings (control and developed product with oil by-product) of olives stuffed with anchovy, red pepper and lemon.

| Ingredients | Stuffing of Anchovy Control | Stuffing of Anchovy with Oil By-Product | Stuffing of Peper Control | Stuffing of Pepper with Oil By-Product | Stuffing of Lemon Control | Stuffing of Lemon with Oil By-Product |
|---|---|---|---|---|---|---|
| Water | 88.00% | 12.00% | 84.90% | - | 88.10% | - |
| Oil | - | 63.33% | - | 63.33 % | - | 63.00% |
| Anchovy paste | 9.00% | 13.00% | - | - | - | - |
| Pepper paste | - | - | 12.10% | 31.00 % | - | - |
| Lemon paste | - | - | - | - | 8.80% | 33.00% |
| Sodium alginate | 3.00% | - | 2.80% | - | 2.90% | - |
| Gelatin | - | 3.33% | - | 4.00 % | - | 3.80% |
| Guar gum | - | - | 0.20% | - | 0.20% | 0.20% |
| Citric acid | - | 8.33% | - | 2.00% | - | - |

To prepare the stuffings of different flavours, the following common procedure was carried out (Supplementary Figure S2): First, the ingredients were weighed according to each of the formulations (Table 1); water, oil by-product, paste of different flavours, water, gelling agents (sodium alginate, gelatin, or guar gum), and citric acid concentrated at 3%. Secondly, the anchovy, pepper, and lemon paste with the rest of the ingredients were mixed and shaken vigorously and mechanically for 1 min using a mechanical homogenizer (Ultraturrax, T18 basic, IKA, Berlin, Germany) until achieve a complete homogenization of the emulsions or stuffings. Finally, those samples recently made were stored at cold temperature [$4 \pm 1\ °C$, 85% of relative humidity (RH)]. Once the different formulations of filling pastes were elaborated (controls and new emulsions made with the oil by-product), the olives were pitted and stuffed by the company (FRUYPER S.A.; Murcia, Spain), by injecting them with the emulsions formulated in Table 1.

On the other hand, different pickled vegetable products were also made from saline concentrates obtained with the brine as a by-product of the same olive pitting and stuffing line. The pickles were made using three vegetable products: carrot, cauliflower, and onion. These products were selected based on their properties or technological capacity for pickling, as well as the external colour of the vegetable (from a range white to orange), which we expected would match the characteristic colour of the brine by-product (orange/brown), without changing the natural colour of the final product. For the initiation of the fermentation process, the brine was concentrated up to a 10% salt concentration (Supplementary Figure S3).

To obtain the three different pickled vegetable products, raw onion, carrot, and cauliflower were received and pre-treated; carrots and onions were manual peeled; and the florets of cauliflower were separated in individual pieces (Supplementary Figure S3). After this process, the three vegetables were washed, cut and manually selected based on their homogenous size and the absence of visual defects. Before the fermentation process, the pre-treated vegetables were weighed and deposited in fermenters of 2.5 L of capacity. The fermentation started with the addition of the brine by-product, which had 2% salt, and it was concentrated to 10% with non-iodized commercial salt for 10 days at 20 °C (75% RH). The 10% salt concentration was maintained at constant levels during the first days of fermentation (approximately one week), with the concentration measured and corrected daily. The concentration of salt in the brine was measured daily by means of a Baumé hydrometer: 1° Baumé (°Bé) corresponded to a 1% salt concentration. On the other hand, the pH of the solution was also measured and corrected daily in the brine by adding vinegar or acetic acid concentrated at 6% of acidity to be constantly at a pH of 4.00



or lower and thus avoid the growth of *Clostridium botulinum* at pHs equal to or greater than 4.50. Once the fermentation process was finished, the product reached an acidity between 0.8 and 2.0%, expressed as lactic acid. Pickled products were washed with clean water to remove the remains of the salt impregnated in the vegetable. Finally, the products were packaged in glass jars, previously sterilized in an autoclave at 120 °C for 10 min, and the covering liquid composed by water acidified with vinegar at 6% with a pH of 3.00, and previously pasteurized at 80 °C for 20 min, was added. After adding the covering liquid, the jars were quickly closed and turned upside down to create a vacuum in the jars. Finally, the new pickled products were cooled with ice.

## 2.2. Fatty Acid Profile

For the characterization of the oil by-product, the fatty acid methyl esters profile (FAME) was identified and quantified (g $L^{-1}$) using a gas chromatography (C-17A; Shimadzu Corporation, Kyoto, Japan) coupled to a flame ionization detector (FID) according to Sánchez- Rodríguez et al. [29]. FAME was prepared with the addition of 13:0 fatty acid as internal standard (0.04 mg $mL^{-1}$). The capillary column used was CPSil-88 (100 m × 0.25 mm ID. 0.2 μm film thickness; J&W 112-88A7; Agilent Technologies, Santa Clara, CA, USA), which is appropriate for olive oil fatty acids separation. Detector temperature was 260 °C, and oils were injected with a 1:20 split ratio. The oven temperature was 175 °C for 10 min, then raised to 220 °C (3 °C $min^{-1}$) and kept at 220 °C for 5 min. The carrier gas was helium, and detector gases were hydrogen (30 mL $min^{-1}$) and air (350 mL $min^{-1}$), and helium (30 mL $min^{-1}$) was used as a make-up gas. Standard solutions (FAME 37 MIX, Supelco; Bellefonte, PA, USA) were injected under the same conditions as oils for the identification of compounds.

## 2.3. Water Activity, pH and Electrical Conductivity and Sterilization of the Brine

Physico-chemical characterization of the brine by-product was carried out through the analysis of water activity (aw), pH and electrical conductivity. The water activity (aw) of the samples was determined using a Novasina equipment (AW SPRINT TH-500, Pedak Meettechniek, Heythuysen, The Netherlands) at 25 °C. The pH analysis was carried out with a pH-meter (CRISON Basic 20, Crison Instruments, Barcelona, Spain) with a sensitivity of ± 0.01 pH. In addition, the electrical conductivity of the brine was measured by a conductivity meter (COND51$^+$, XS Instruments, Carpi, Italy) and it was expressed as mS $cm^{-1}$ [30]. Finally, the brine was sterilized at 120 °C for 10 min by using an autoclave Stericlasv-S (Trade Raypa®, Terrasa, Spain) to ensure the food safety of this by-product.

## 2.4. Total Phenolics

Total phenolics of both by-products (oil, and autoclaved and non-autoclaved brine) as well as of the different new products developed (anchovy, red pepper, and lemon pastes; pickled carrot, cauliflower and onion) were analysed. Phenolic extraction was performed using 5 g of the product homogenized with 15 mL of methanol: water (80:20, *v/v*) containing 2 mM of sodium fluoride, for 2 min by using a homogenizer (IKA T18 basic Ultraturrax, Deutschland, Germany). The determination of total phenolics was carried out using the Folin–Ciocalteu method according to the protocol described by Wood et al. [31]. The absorbance at 760 nm was read in duplicate for each extract using a spectrophotometer (UV-1700 PharmaSpec; Shimadzu Corporation, Kyoto, Japan). Results were expressed as mg equivalents of gallic acid per 100 g or mL of sample, and the analyses were performed in duplicate for each sample.

## 2.5. Microbiological Analysis

To ensure the food safety of the brine by-product, different microbiological determinations were carried out: mesophilic aerobic count, moulds, and yeasts, enterobacteria, *Escherichia coli*, *Salmonella* spp., *Listeria* spp. and *Listeria monocytogenes*. The counts of mesophilic aerobic, and moulds and yeasts were performed using Petrifilm (3M) plates

and VRBD Agar medium, respectively, and expressed in Colony Forming Units per gram of product (CFU g$^{-1}$). However, the rest of the counts were based on absence or presence and/or on a positive (+) or negative (−) count. The Hygicult E/β-GUR test (AIDIAN) was used for the detection of enterobacteria and β-glucuronidase-producing species, including *E. coli*. Analyses for *Salmonella* species were performed by an external laboratory. Finally, the count of *Listeria* species and, specifically, *Listeria monocytogenes* was carried out through the colour change test using InSite$^{TM}$ L. mono Glo (certified by the AOAC (Association of Analytical Communities) International).

### 2.6. Sensorial Analysis

Finally, a descriptive sensory analysis of all the new products developed was carried out with a trained panel of 10 judges (from 25 to 55 years old; 50% men and 50% women), with more than 1000 h of experience in sensory analysis of food from the Agro-food Technology Department from Miguel Hernández de Elche University (Orihuela, Alicante, Spain). Two preliminary orientation sessions were realized to discuss the most valued attributes in these products by consumers. Specifically, the sensory attributes evaluated in the stuffed olives were emulsion colour, ID aroma (anchovy, pepper, or lemon), olive aroma, unctuousness in mouth, ID taste (anchovy, pepper, or lemon), olive taste, salty, acidity, olive aftertaste, and overall liking. In addition, colour, colour uniformity, size, hardness, crispness, acidity, salty, aftertaste, brine colour, brine acidity and overall liking were described in the new pickled vegetables. The panelists used a numerical scale for the intensity of each corresponding attribute: 0 (no intensity) to 10 (extremely strong), with changes of 0.5 units. In the case of attributes related to the homogeneity, a value of 5.00 would be a homogeneous attribute in the studied product, according to the procedure described by Cano-Lamadrid et al. [32] in table olives. The control olives were those made by the commercial company with the sodium alginate gelling agent, while the control vegetable pickled were commercial products from the market. Values of 5.00 were attributed to all of them in all the evaluated attributes as reference.

### 2.7. Statistical Analysis

Statistical analysis was performed using five replicates for all analytical determinations. Results were expressed as the mean ± SE. The data were submitted to an analysis of variance (ANOVA). Mean comparisons were made using Student's *t*-test for two data subsets and Tukey's test for a larger number of samples, in order to determine if the differences between the samples were significant at $p < 0.05$ in both tests. The significant differences have been indicated using symbol (*) (comparison between control and oil-emulsified pastes or among fresh and brine-pickled vegetable) and lower-case letters (comparison of the olive by-product with the new products made with it). All statistical analyses were performed using the SPSS v.17.0 software package for Windows.

## 3. Results

### 3.1. Microbiological Traits of Olive Wastewater as Raw Material

Microbiological load of aerobic mesophilic and moulds and yeasts were $2 \cdot 10^3$ and $5 \cdot 10^3$ CFU g$^{-1}$, respectively, of brine by-product (Supplementary Table S1). Non-autoclaved sample was not contaminated by enterobacteria and *E. coli*, although *Listeria* spp. were detected in samples. Nevertheless, the presence of *Listeria monocytogenes*, which is the only one involved in effects on human health, was negative in the brine by-product (Supplementary Table S1).

### 3.2. Characterization of Oil and Brine Generated as Olive By-Products

Olive oil and brine by-products were generated from the wastewater of the olive pitting process, and, after the centrifugation process, which was detailed above. Both by-products were characterized to determine their suitability to be used as co-products for value-added food development. Specifically, the fatty acid profile and content of olive oil

were analysed to be incorporated as an emulsifier in the olive stuffing, as can be observed in Figure 2. Three oil-emulsified pastes were developed: anchovy, red pepper, and lemon. On the other hand, the brine by-product was used to pickle three types of vegetables: carrot, cauliflower, and onion. Previously, the brine was characterized in terms of its physico-chemical and functional parameters (Table 2).

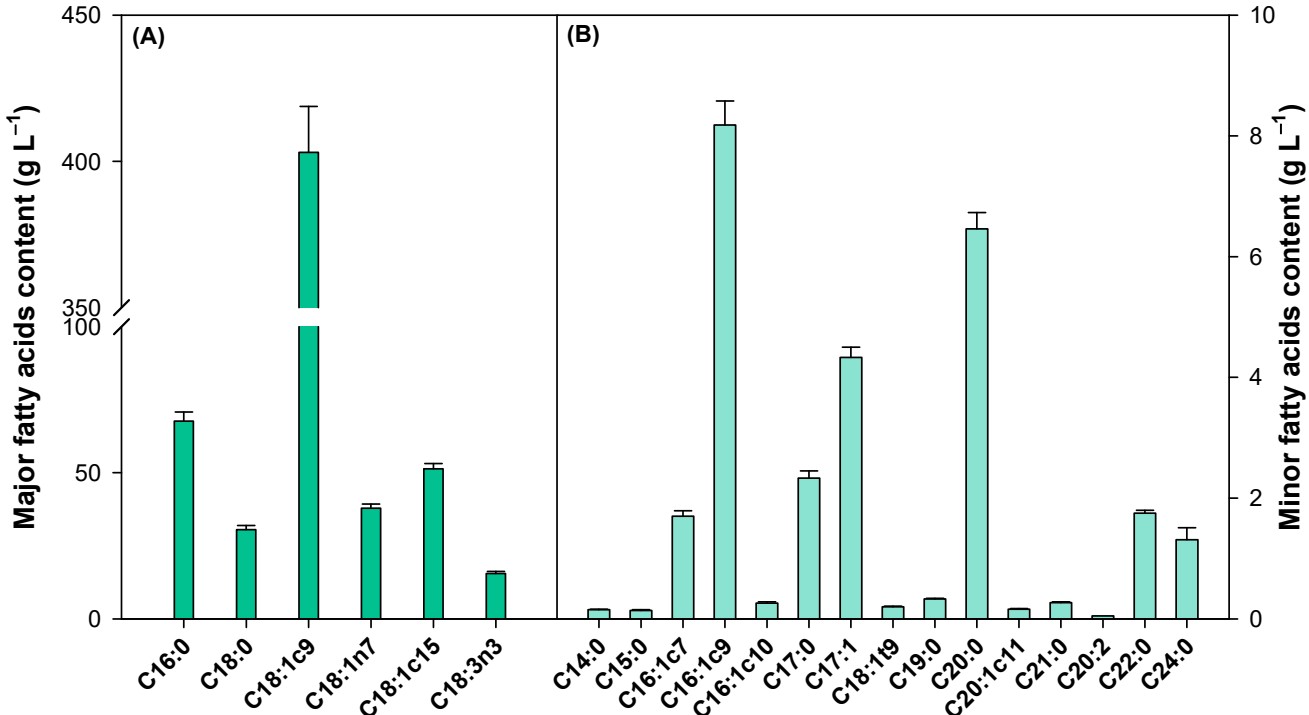

**Figure 2.** Major (**A**) and minor (**B**) fatty acid content of olive oil generated as olive by-product.

**Table 2.** Physico-chemical and functional parameters of brine generated as olive by-product.

| pH | Conductivity (mS cm$^{-1}$) | a$_W$ | Total Phenolics (mg 100 mL$^{-1}$) Non-Autoclaved Sample [†] | Total Phenolics (mg 100 mL$^{-1}$) Autoclaved Sample [†] |
|---|---|---|---|---|
| 4.08 ± 0.01 | 14.41 ± 0.02 | 0.80 ± 0.02 | 16.56 ± 0.10 NS | 16.39 ± 0.07 NS |

[†] NS: No significant differences ($p \geq 0.05$) between non-autoclaved and autoclaved samples.

### 3.2.1. Fatty Acid Profile and Content of Olive Oil By-Product

The composition of the olive oil by-product is presented in Figure 2. Twenty-one fatty acids were identified and quantified in the oil, the major one being C18:1c9 (oleic) acid (62.62%), followed by C16:0 (palmitic), C18:1c15 (−), C18:1n7 (*cis*-vaccenic), C18:0 (stearic) and C18:3n3 (linolenic) acids (Figure 2A). In addition, other fifteen minor fatty acids were also identified and quantified in lower concentrations: C16:1c9 (palmitoleic), C20:0 (arachidic), C17:1 (margaroleic), C17:0 (margaric), C22:0 (behenic), C16:1c7 (palmitoleic), C24:0 (lignoceric), C19:0 (nodecanoic), C21:0 (heneicosylic), C16:1c10 (*cis*-heptadecenoic), C18:1t9 (elaidic), C20:1c11 (gadoleic), C14:0 (myristic), C15:0 (pentadecanoic) and C20:2 (*cis*-11,14 eicosadienoic) acids, ordered from higher to lower proportion with respect to the composition of the oil (Figure 2B). Olive oil by-product was rich in unsaturated fatty acid (82.77%), monounsaturated being the major ones (80.34%), while the saturated fatty acids fraction represented 17.23% of the total profile.

### 3.2.2. Physico-Chemical and Functional of Brine By-Product

The physico-chemical parameters analysed in the brine fraction were pH, conductivity and a$_W$ (Table 2). The pH and conductivity values were ≈ 4.08 and 14.41 mS cm$^{-1}$,

respectively. The brine showed an a$_W$ value of $0.80 \pm 0.02$, as can be observed in Table 2. On the other hand, functional characterization of the non-autoclaved brine taken in situ from the centrifuges directly during the pitting stage of the olive was performed. A great concentration of total phenolic compounds, up to $16.56 \pm 0.10$ mg 100 mL$^{-1}$, was observed in the brine by-product. The total phenolics content did not show significant differences ($p \geq 0.05$) between non-autoclaved and autoclaved samples (Table 2).

### 3.3. Use of Oil By-Product as Co-Product to Develop Stuffing for Olives

3.3.1. Functional Characterization of Olive Oil By-Product and Control and Oil-Emulsified Pastes (Anchovy, Red Pepper, and Lemon)

The olive oil by-product and control and oil-emulsified pastes were characterized in terms of total phenolics content (Figure 3). This bioactive content was significantly ($p < 0.05$) improved in the three oil-emulsified pastes compared to control ones, being the increase in phenolics contributed by the oil by-product addition. The olive oil by-product showed the lowest total phenolic content (Figure 3A), followed by the oil-emulsified anchovy paste (Figure 3B). In this sense, the emulsions based on vegetables (red pepper and lemon fruit) presented the highest content of bioactive compounds, although no significant differences ($p \geq 0.05$) were observed between both (Figure 3C,D).

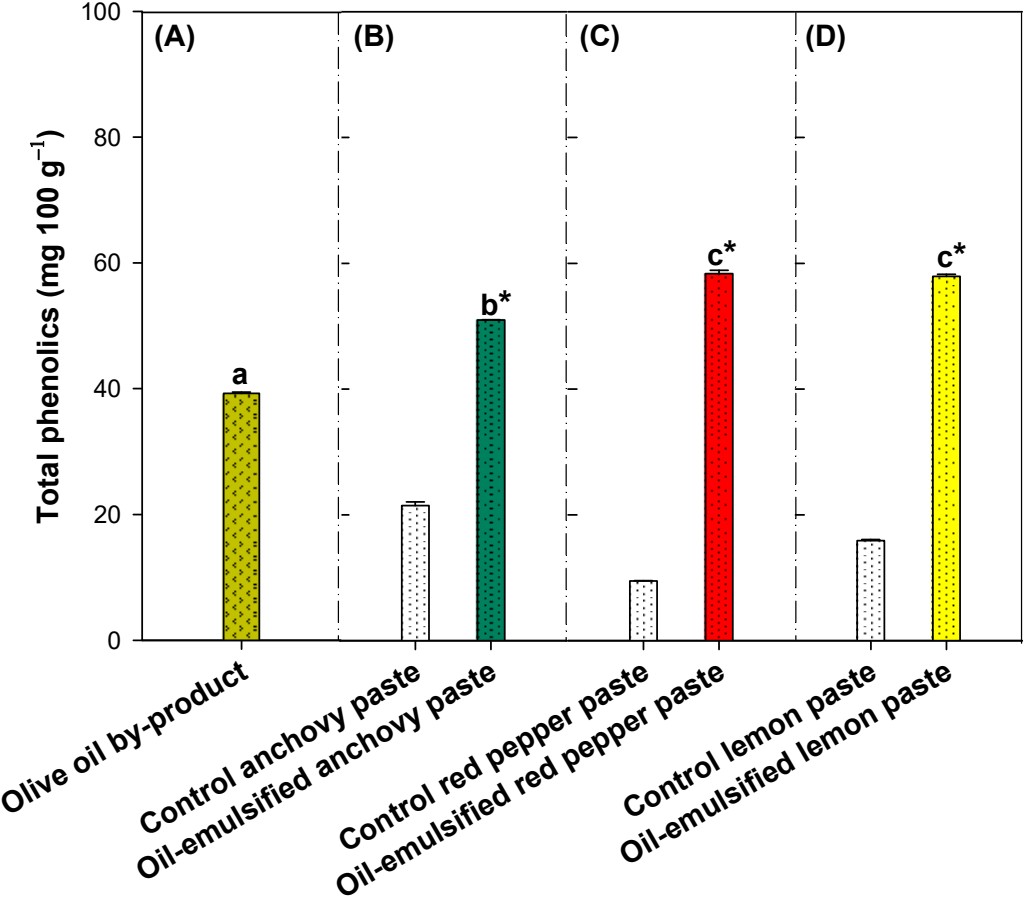

**Figure 3.** Total phenolics content of olive oil by-product (**A**); and control and oil-emulsified pastes [anchovy (**B**); red pepper (**C**); and lemon (**D**)]. Symbol (*) shows significant differences between control and oil-emulsified pastes at $p < 0.05$ for each type of emulsion (anchovy, red pepper, and lemon). Lowercase letters show significant differences at $p < 0.05$ among olive oil by-product and the three oil-emulsified pastes (anchovy, red pepper, and lemon).

### 3.3.2. Sensory Analysis of Control and Oil-Emulsified Pastes (Anchovy, Red Pepper, and Lemon)

The three new types of stuffed olives (anchovy, red pepper, and lemon) developed were sensory compared with control olives emulsified with sodium alginate. The parameters evaluated were emulsion colour, ID aroma, olive aroma, unctuousness in mouth, ID taste, olive taste, salty, acidity, olive aftertaste, and overall liking (Table 3). The oil-emulsified anchovy paste showed a significantly lower ($p < 0.001$) emulsion colour than the control, although its values of ID aroma (anchovy), unctuousness in mouth, ID taste (anchovy), salty, olive aftertaste, and overall liking were significantly higher ($p < 0.001$). The red pepper paste emulsified with the olive oil by-product presented a significantly higher ($p < 0.001$) ID aroma (red pepper fruit), unctuousness in mouth, ID taste (red pepper fruit), olive aftertaste and overall liking compared to control red pepper paste, while the emulsion colour was significantly lower ($p < 0.001$). Finally, all sensory attributes studied of the oil-emulsified lemon paste, except olive aroma, olive taste and salty, were valued above the control score ($5.00 \pm 0.00$), showing significant differences ($p < 0.001$) with respect to control red pepper paste (Table 3).

**Table 3.** Sensory analysis of stuffed olives: control and oil-emulsified pastes (anchovy, red pepper, and lemon).

| Sensory Attribute | ANOVA [†] | Control | Anchovy | ANOVA [†] | Control | Red Pepper | ANOVA [†] | Control | Lemon |
|---|---|---|---|---|---|---|---|---|---|
| Emulsion colour | *** | $5.00 \pm 0.00$ | $4.00 \pm 0.07$ | *** | $5.00 \pm 0.00$ | $4.00 \pm 0.07$ | *** | $5.00 \pm 0.00$ | $7.00 \pm 0.07$ |
| ID aroma: anchovy, pepper, or lemon | *** | $5.00 \pm 0.00$ | $7.00 \pm 0.11$ | *** | $5.00 \pm 0.00$ | $6.00 \pm 0.11$ | *** | $5.00 \pm 0.00$ | $6.00 \pm 0.11$ |
| Olive aroma | NS | $5.00 \pm 0.00$ | $5.00 \pm 0.07$ | NS | $5.00 \pm 0.00$ | $5.00 \pm 0.07$ | NS | $5.00 \pm 0.00$ | $5.00 \pm 0.07$ |
| Unctuousness in mouth | *** | $5.00 \pm 0.00$ | $7.00 \pm 0.07$ | *** | $5.00 \pm 0.00$ | $7.00 \pm 0.07$ | *** | $5.00 \pm 0.00$ | $8.00 \pm 0.07$ |
| ID taste: anchovy, pepper, or lemon | *** | $5.00 \pm 0.00$ | $7.00 \pm 0.07$ | *** | $5.00 \pm 0.00$ | $7.00 \pm 0.07$ | *** | $5.00 \pm 0.00$ | $8.00 \pm 0.07$ |
| Olive taste | NS | $5.00 \pm 0.00$ | $5.00 \pm 0.16$ | NS | $5.00 \pm 0.00$ | $5.00 \pm 0.16$ | NS | $5.00 \pm 0.00$ | $5.00 \pm 0.16$ |
| Salty | *** | $5.00 \pm 0.00$ | $6.00 \pm 0.07$ | NS | $5.00 \pm 0.00$ | $5.00 \pm 0.14$ | NS | $5.00 \pm 0.00$ | $5.00 \pm 0.14$ |
| Acidity | NS | $5.00 \pm 0.00$ | $5.00 \pm 0.07$ | NS | $5.00 \pm 0.00$ | $5.00 \pm 0.07$ | *** | $5.00 \pm 0.00$ | $7.00 \pm 0.07$ |
| Olive aftertaste | *** | $5.00 \pm 0.00$ | $6.00 \pm 0.07$ | *** | $5.00 \pm 0.00$ | $7.00 \pm 0.07$ | *** | $5.00 \pm 0.00$ | $7.00 \pm 0.07$ |
| Overall liking | *** | $5.00 \pm 0.00$ | $7.00 \pm 0.07$ | *** | $5.00 \pm 0.00$ | $7.00 \pm 0.07$ | *** | $5.00 \pm 0.00$ | $8.00 \pm 0.14$ |

[†] NS: No significant ($p \geq 0.05$). Symbol of ***, within the same type of stuffed olive, denote significant differences at $p < 0.001$.

### 3.4. Use of Brine as Co-Product to Develop Pickled Vegetables

3.4.1. Functional Characterization of Brine By-Product and Fresh and Pickled Vegetables (Carrot, Cauliflower, and Onion)

Among the three types of pickled vegetables, all samples pickled with the brine by-product showed a significant higher ($p < 0.05$) content than fresh vegetables, being the brine by-product the sample with the lowest content (Figure 4A). On the contrary, brine-pickled onion showed the highest total phenolics content (Figure 4D), followed by cauliflower and carrot pickles (Figure 4B,C).

3.4.2. Sensory Analysis of Fresh and Pickled Vegetables (Carrot, Cauliflower, and Onion)

Table 4 shows the sensory attributes evaluated in control commercial products and the new brine-pickled vegetables (carrot, cauliflower, and onion). The brine-pickled carrot had a significantly lower ($p < 0.001$) colour and brine acidity than control pickle. Nevertheless, those pickled carrots showed a significantly higher ($p < 0.001$) size, hardness, crispness, salty, aftertaste and overall liking than controls. In addition, brine-pickled cauliflower showed a significantly higher ($p < 0.001$) hardness, crispness, salty, aftertaste, and overall liking compared to the control, although its colour, size and brine acidity values were significantly lower ($p < 0.001$).

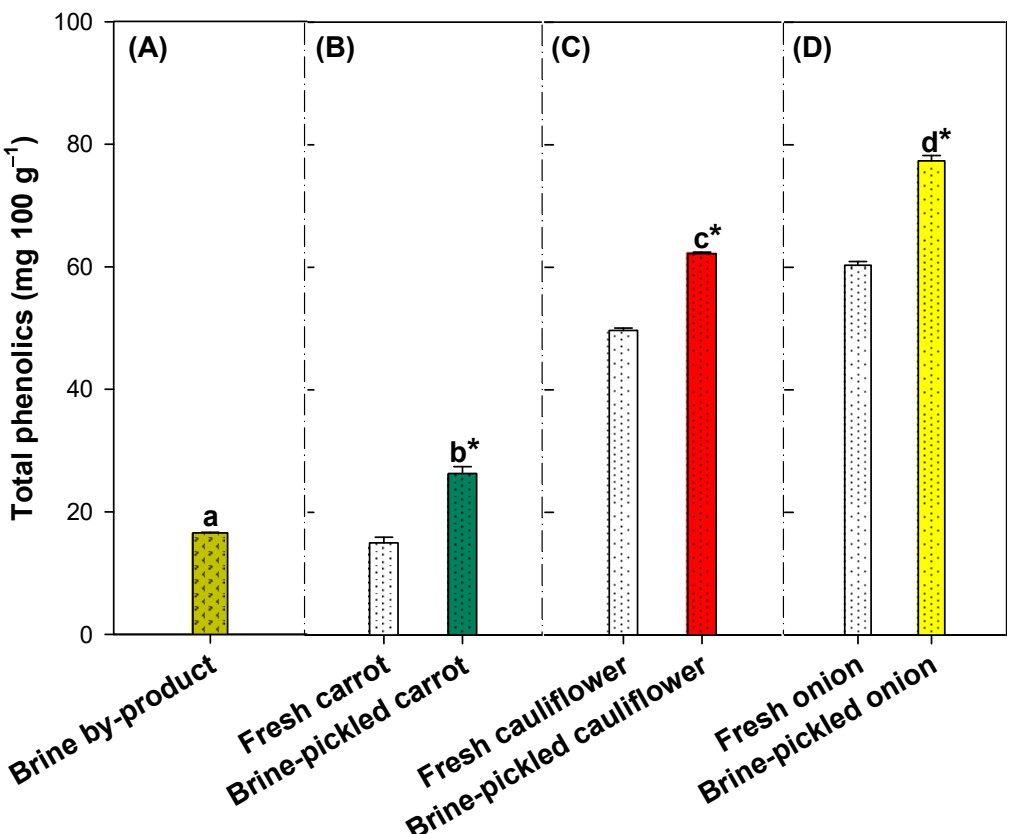

**Figure 4.** Total phenolics content of brine by-product (**A**); and fresh and pickled vegetables [carrot (**B**); cauliflower (**C**); and onion (**D**)]. Symbol (*) shows significant differences between fresh and brine-pickled vegetable at $p < 0.05$ for each type (carrot, cauliflower, and onion). Lowercase letters show significant differences at $p < 0.05$ among brine by-product and the three brine-pickled vegetables (carrot, cauliflower, and onion).

**Table 4.** Sensory analysis of control and brine-pickled vegetables (carrot, cauliflower, and onion).

| Sensory Attribute | ANOVA [†] | Control | Carrot | ANOVA [†] | Control | Cauliflower | ANOVA [†] | Control | Onion |
|---|---|---|---|---|---|---|---|---|---|
| Colour | *** | 5.00 ± 0.00 | 4.00 ± 0.07 | *** | 5.00 ± 0.00 | 2.50 ± 0.11 | *** | 5.00 ± 0.00 | 7.00 ± 0.07 |
| Colour uniformity | NS | 5.00 ± 0.00 | 5.00 ± 0.00 | NS | 5.00 ± 0.00 | 5.00 ± 0.00 | *** | 5.00 ± 0.00 | 3.50 ± 0.13 |
| Size | *** | 5.00 ± 0.00 | 6.00 ± 0.07 | *** | 5.00 ± 0.00 | 4.00 ± 0.11 | *** | 5.00 ± 0.00 | 4.00 ± 0.11 |
| Hardness | *** | 5.00 ± 0.00 | 6.00 ± 0.00 | *** | 5.00 ± 0.00 | 6.00 ± 0.00 | *** | 5.00 ± 0.00 | 6.00 ± 0.00 |
| Crispness | *** | 5.00 ± 0.00 | 7.00 ± 0.05 | *** | 5.00 ± 0.00 | 7.00 ± 0.05 | *** | 5.00 ± 0.00 | 6.00 ± 0.11 |
| Acidity | NS | 5.00 ± 0.00 | 5.00 ± 0.05 | NS | 5.00 ± 0.00 | 5.00 ± 0.05 | *** | 5.00 ± 0.00 | 6.00 ± 0.14 |
| Salty | *** | 5.00 ± 0.00 | 6.00 ± 0.07 | *** | 5.00 ± 0.00 | 6.00 ± 0.07 | *** | 5.00 ± 0.00 | 6.00 ± 0.07 |
| Aftertaste | *** | 5.00 ± 0.00 | 7.00 ± 0.05 | *** | 5.00 ± 0.00 | 8.00 ± 0.11 | *** | 5.00 ± 0.00 | 7.00 ± 0.05 |
| Brine colour | NS | 5.00 ± 0.00 | 5.00 ± 0.05 | NS | 5.00 ± 0.00 | 5.00 ± 0.05 | NS | 5.00 ± 0.00 | 5.00 ± 0.05 |
| Brine acidity | *** | 5.00 ± 0.00 | 3.50 ± 0.09 | *** | 5.00 ± 0.00 | 3.50 ± 0.09 | *** | 5.00 ± 0.00 | 3.50 ± 0.09 |
| Overall liking | *** | 5.00 ± 0.00 | 7.00 ± 0.07 | *** | 5.00 ± 0.00 | 8.00 ± 0.11 | *** | 5.00 ± 0.00 | 7.00 ± 0.11 |

[†] NS: No significant ($p \geq 0.05$). Symbol of ***, within the same type of pickled vegetables, denote significant differences $p < 0.001$.

Finally, the brine-pickled onion showed significant differences ($p < 0.001$) in all the sensory attributes, except on brine colour, compared to control product. Specifically, colour, hardness, crispness, acidity, salty, aftertaste and overall liking were significantly higher ($p < 0.001$) than control reference. On the contrary, colour uniformity, size and brine acidity values were significantly lower ($p < 0.001$) than the reference ones (Table 4).

## 4. Discussion

### 4.1. Olive Oil and Brine as Potential By-Products to Develop Value-Added Foods

In the last few years, consumers have become more aware of the ingredients in food and have paid more attention to the valorization of agro-industrial waste and to the utilization of by-product, promoting their re-use to develop new functional foods. However, in addition to the fact that the by-products obtained must present an optimal technological function, they must comply with the criteria of quality and food safety. The microbial community present in the olive wastewater is strongly influenced by several parameters, among which the ripeness state and the olive cultivar are the most influential [33,34]. The microbiological characterization of the wastewater showed that the count of mesophilic aerobes and moulds and yeasts were in an acceptable range or considered as a solution with a low and apt microbial load, mainly determined by facultative microorganisms and anaerobic strata originally present in the plant fresh, for use as a co-product. These microbial densities of wastewater were lower than the values reported by other authors [33–36], which were mainly composed of yeasts, bacteria, and moulds. However, the sample used in this experiment was autoclaved (Supplementary Table S1).

Vegetable by-products are considered valuable sources for the formulation of new value-added food products [37]. The olive oil was separated from wastewater by centrifugation and its fatty acid profile showed that this by-product was rich in monounsaturated fatty acid (80.34%), the major one being C18:1c9 (oleic) acid (62.62%), followed by C16:0 (palmitic), C18:1c15 (−), C18:1n7 (*cis*-vaccenic), C18:0 (stearic) and C18:3n3 (linolenic) acids (Figure 2A). Accordingly, other studies have reported that the lipid fraction of oil pomace is particularly rich in oleic acid (*ca.* 75%), followed by palmitic, linoleic, and stearic acids [13,38–40]. Moreover, the same fatty acid profile was reported in other olive cultivars; 'Arbequina' [41] and 'Picual' [42], as well as in 'Hojiblanca' [43], which was the cultivar used in the present study. The fatty acid composition of oil differs depending on the plant species and, in the case of the olive tree, there could be also differences among cultivars, maturity stage and production area [41]. Values for the minor saturated C14:0, C15:0, C17:0, C20:0, C22:0, and C24:0 fatty acids were lower compared with the majors saturated fatty acids identified: C16:0 and C18:0. With respect to C16:1c7, C16:1c9, C16:1c10, C17:1, C18:1t9, C20:1c11, and C20:2, the unsaturated fatty acids were also detected in lower contents than C18:1c9, C18:1n7, C18:1c15 and C18:3n3. However, these minor values were in similar proportions to those reported by other authors in olive oil [41–43]; thus, no differences were observed on fatty acid profile related to the extraction process carried out in the present experiment.

Oleic acid is a fatty acid of special interest since unsaturated fatty acids are known to be better absorbed than saturated ones [44], apart from the consumption of saturated fat has been related to many health concerns [45,46]. It should be highlighted that the European Union (EU) published Regulation (EU) No. 432/2012 of the Commission [47] containing the list of 222 health claims authorized for commercial and advertising use. Among the components included in this list, it appears, in addition to water, fiber or certain vitamins, oil. Specifically, the oleic acid is declared as unsaturated fatty acid that would contribute on the diet to maintain normal levels of blood cholesterol. Therefore, it could be concluded that the oil obtained as a by-product of the pitting line of stuffed olives could be considered as a healthy nutrient which would provide a value-added as a co-product to the final food product. However, all the approved health claims regarding oleic acid will be related to its quantity and, specifically, the amount of oil that the developed food product will include in its formulation, being a high percentage (~63%) (Table 1).

The brine generated as olive by-product showed a pH value of 4.08 (Table 2), according to other studies carried out with brines from the processing of olives that showed values of pH close to 4.00 [48]. On the other hand, the conductivity value was 14.41 mS cm$^{-1}$ which means that it is a solution concentrated in salt (Table 2). The pH value of the brine by-product agreed with the range of 2.2–5.9 pH published to characterize the oil mill wastewater by Demerche et al. [14]. However, the electrical conductivity was higher in

the present study than the range of values of 5.5–10 mS cm$^{-1}$ reported in other study [14]. Regarding the relation between electrical conductivity and salinity of the solution, the conductivity value obtained presented approximately a salt content of 1.8% (180 g of NaCl dissolved in 820 mL of H$_2$O; calculations based on own calibration curves that relate electrical conductivity with estimated NaCl concentration data not shown). Finally, olive brine showed an a$_W$ value of 0.803 (Table 2). The osmotic removal of water from food reduces the water activity to a level according to the amount of salt added. In this sense, Robinson and Stokes [49] related the a$_W$ value of a salt solution with the NaCl content of the solution and, based on this relation, the a$_W$ value of the present study would correspond to 30.10 g of NaCl for every 100 mL of H$_2$O. This salt content remains similar to that calculated with electrical conductivity calibration curves, but it was less effective. However, the initial salt % of brine was determined by using the Baumé salinometer, which confirmed that this by-product was concentrated to 1.8% (1.8 °Bé value).

All pH, electrical conductivity and a$_W$ values obtained after physico-chemical characterization of brine confirm that it is an olive by-product with parameters acceptable for its use as a co-product in the development of value-added foods. In addition, its characteristics related to a$_W$ value led to predict microbial safety in the by-product. According to Lee [50] and Leistner et al. [51], a food product with a maximum value of a$_W$ of 0.81 could only grow yeasts and fungi of the *Saccharomyces* and *Penicillium* genus, respectively. At the functional level, the phenolic content was quantified in the brine by-product (Figure 2), highlighting that there was no significant difference ($p \geq 0.05$) with respect to the bioactive content after sterilization at 120 °C for 10 min. This result was favorable regarding its use as a co-product since the main aim of the present study is add it functional compounds, mainly these phenolic compounds from the olive after being pitted, to the final product developed. These results were not in agreement with other studies that suggested that phenolic are unstable to heat and this is why the polyphenols were lower than those predicted [52,53]. Vogrinčič et al. [54] showed a decrease in polyphenols concentrations (rutin, quercetin and polyphenols) of tartary buckwheat bread as result of baking. Verardo et al. [55] also studied the effects of the pasta making and boiling processes, showing a decrease of ~ 53% of total phenolic compounds in the cooked spaghetti, due to the solubility of phenolic compounds in the cooking water. However, Abdel-Aal and Rabalski [56] suggested that the decrease in bioactive compounds depends on the type of products, on the recipe and processing conditions but mainly on the type of phenolic compounds.

Antioxidants, including phenolic compounds, are different in terms of thermal stability. Many of the antioxidants that exhibit high preventive activity at storage temperatures break down quickly and lose their effectiveness when they are exposed to elevated temperatures. Furthermore, during certain industrial processing, antioxidants gradually escape from the environment through the steam generated during the process. As a result, thermal decomposition and leaching or loss of antioxidants are the two main factors that reduce its effectiveness at high temperatures [57]. Elhamirad and Zamanipoor [58] studied the thermal stability and drag properties of these functional compounds under thermal decomposition and leaching conditions. In that study, the authors observed that the thermal stability of some phenolic compounds or antioxidants, such as quercetin and ellagic acid, was very high [58]. In another study, it was observed that during the kinetics of thermal decomposition the molecule or active principle hydroxytyrosol-OLPD with anti-inflammatory capacity showed that for temperatures around 100 °C it retained its integrity and stability. Nevertheless, when it was subjected to conditions between 170 and 220 °C for 150 min, it decomposed [59]. Therefore, results about the absence of thermal degradations of the phenolic compounds present in the olive by-product could be explained from the point of view of thermodynamic parameters and kinetics, in that these active principles are related to temperature/time ratio and to thermal process around 100 °C, although the individual phenolic profile of the olive by-product requires furter research. Finally, the physico-chemical and functional characteristics of the olive oil and brine generated as

by-products of the olive industry showed a high technological and functional capacity for their reuse as co-products.

### 4.2. Olive Oil By-Product as Potential Co-Product to Develop Stuffing for Olives

As far as we know, this is the first article which uses the olive oil by-product presented in the wastewater generated from the pitting process of the olive to develop different stuffing types for olives. Anchovy, red pepper, and lemon stuffings emulsified with the olive oil by-product were developed and characterized in terms of total phenolics content (Figure 3). Total phenolic content of the three different stuffings developed with the olive oil by-product was significantly ($p < 0.05$) higher than the content of oil by-product and control emulsions. These differences regarding the phenolic content between the control formulations and the new ones developed could be determined by the influence of the total phenolic content of the olive oil by-product used to develop the emulsions. Various studies have confirmed that olive oil has a high phenolic content [60,61], which is added to the total phenolic content presented in each of the pastes (anchovy, red pepper, and lemon), giving the final product an added functional value. Among the emulsions with the highest phenolic content are those developed from pepper and lemon pastes, since they were elaborated from a raw vegetable material with a high polyphenolic content [62,63], compared to the possible polyphenolic content present in anchovy paste. Therefore, this functional improvement in the new emulsions developed could be related, at least in part, to the health claim relating to the polyphenolic content of olive oil that declares that: a minimum content of 5 mg of hydroxytyrosol and its derivatives per 20 g of olive oil contribute to the protection of blood lipids from oxidative stress [47]. Hydroxytyrosol is widely present in olive oil by-products, and its antioxidant and antimicrobial properties have been widely demonstrated [15,64,65]. However, to fully support this claim, it would be necessary to quantify the content of hydroxytyrosol present in the olive obtained as by-product, which would be interesting as future research line since antioxidants and phenolic compounds play an important role in the healthy properties of the olive oil [66].

Table 3 shows a sensory analysis of stuffed olives with the new emulsions developed compared to control formulations. In this sense, olives stuffed with anchovy presented significant differences ($p < 0.05$) in the attributes of anchovy odour, unctuousness in mouth, anchovy flavour, salty, olive aftertaste and overall liking, these being evaluated positively than control. However, the colour attribute of the new anchovy emulsion elaborated with the oil by-product showed a significantly ($p < 0.05$) lower value than control. Colour of emulsions could be influenced by the addition of oil in the formulation or by reaching protein denaturation after achieving the isoelectric point of anchovy proteins by adding citric acid in the new formulation (Table 1). Similarly, the addition of oil as an emulsifier to the new stuffings improved sensory properties of the other two emulsions based on pepper paste and lemon paste (Figure 1). Results also showed significant ($p < 0.05$) differences for both formulations compared to controls in terms of pepper/lemon aroma, unctuousness in mouth, pepper/lemon taste, acidity (in the case of lemon), olive aftertaste and overall liking. Regarding colour, both formulations presented significant differences ($p < 0.05$) compared to controls. Red pepper emulsion showed the same problem regarding colour as anchovy paste since this new emulsion diluted the characteristic reddish colour of red pepper, getting lost (Figure 1). Nevertheless, the new lemon emulsion developed showed an improvement on colour since the yellow tone of this emulsion increased and it seemed to trained judges that it contained more lemon, and it was more natural (Figure 1). The development of new formulations for the preparation of fillings for stuffed olives with anchovy, red pepper, and lemon, using the olive oil as by-product, showed to be a potential strategy to elaborate foods with a value-added and more functional, partially reducing a problem of high environmental impact.

*4.3. Brine By-Product as Potential Co-Product to Develop Pickled Vegetables*

Total phenolic content was quantified in both the brine obtained as a by-product and in fresh and pickled vegetables (Figure 4). After the fermentation, the three pickled vegetables with the brine by-product had a significantly higher phenolic content ($p < 0.05$) than fresh products. Therefore, it could be concluded that phenolic compounds from the by-product were transferred practically in their entirety to the new product developed, giving it value-added. The polyphenolics presented in the brine come from the pitting process of the olive. Among the different vegetables tested, onion, both fresh and pickled, presented the highest polyphenolics concentration. Some studies have reported that the whole onion and industrial onion residues of 'Recas' cultivar was rich on polyphenolics compounds [67,68]. With respect to industrial onion residues, the brown skin presented the highest levels of total polyphenolics and flavonoids, and the inner layers the lowest values. Therefore, there is a decrease in the total phenolic compounds, as well as the total flavonoids, from the outside of the bulb to the inside [68].

Finally, the sensory analysis of the three pickled vegetable products; (A) carrot, (B) cauliflower, and (C) onion, was carried out by comparing with a control or reference acquired at supermarket. As can be seen in Table 4, the pickled carrot was valued more highly than the reference product, standing out in terms of colour, size, hardness, crispness, salty, aftertaste and overall liking. Similarly, the pickled cauliflower with the brine by-product also presented improvements in the same sensory attributes evaluated by the descriptive panel with significant differences ($p < 0.05$) compared to controls in some of them. Specifically, the white colour desired by consumers in the final product and associated with a higher quality was positively highlighted by the panelists. The same attributes in the other vegetables were highlighted in the new pickled onions, although the colour and the colour homogeneity were not valued positively by the trained judges compared to the control. As has been previously commented, this fact could be due to the developed dark colour at the end of the fermentation process which was not homogeneous in all onion cuts. The natural colour loss of the onion could occur through an oxidation process after fermenting in brine, as it was observed previously in the lactic fermentation of table olive [69]. Therefore, it can be concluded that all pickled vegetables showed a higher overall liking by the descriptive panel, the differences being significant with respect to the commercial product or reference ($p < 0.05$). It should be highlighted that the pickled with the highest overall liking among the three vegetables tested was the cauliflower since it improved pleasantly its sensory traits compared to the commercial sample.

## 5. Conclusions and Future Challenges

In conclusion, the reuse and/or valorization of the two by-products generated in the pickled and/or stuffed olive industry; oil (waste) fat and brine (aqueous residue) is possible for the development of value-added food products. Both by-products contributed to the developed food a value-added due to their content in phenolic compounds and the conversion of a by-product into co-product in line with the concept of circular economy since the final product presented very interesting functional (total phenolic content) and sensory (descriptive analysis) properties and were even better than those of the commercial products.

Finally, a transition to circular business models must be made. Since most olive oil producers are expected to reuse by-products in the near future, due to its potential technological and functionality capacity, this research work could be a basis to develop new value-added foods. In this sense, future research using the same conditions of this study in the field are encouraged. Although the potential application of oil mill wastewater for setting up functional food products as a natural concentrate of substances with antioxidant action could be a promising opportunity, to date no reference legislation on the market for the use of these by-products for human consumption is currently available. On the other hand, the by-product separation process is expensive for the industry, which would entail

a limitation in the implementation. However, cost of the treatments could be compensated by the income from useful by-products.

**Supplementary Materials:** The following supporting information can be downloaded at: https://www.mdpi.com/article/10.3390/agronomy13030718/s1, Figure S1: Flow diagram of the olive pitting process from which the olive oil and brine by-products that are the object of study of the present experiment are obtained.; Figure S2: Flow diagram of the development of three oil-emulsified pastes (anchovy, pepper, and lemon). Figure S3: Flow diagram of the development of three pickled vegetables (carrot, onion, and cauliflower) with the brine by-product.; Table S1: Microbiological characteristics of brine by-product.

**Author Contributions:** Conceptualization, M.E.G.-P., M.J.G. and P.J.Z.; methodology, A.D.-S. and M.R.-S.; software, M.E.G.-P.; validation, M.E.G.-P., M.J.G. and P.J.Z.; formal analysis, M.E.G.-P., M.R.-S. and A.D.-S.; investigation, M.E.G.-P., A.D.-S. and M.R.-S.; resources, P.J.Z.; data curation, M.E.G.-P. and M.J.G.; writing—original draft preparation, M.E.G.-P. and M.J.G.; writing—review and editing, M.E.G.-P., M.J.G. and P.J.Z.; supervision, P.J.Z.; project administration, P.J.Z. All authors have read and agreed to the published version of the manuscript.

**Funding:** Part of the analytical equipment used in this study was purchased through the grant EQC2018-004170-P funded by MCIN/AEI/10.13039/501100011033 and by ERDF "A way to make Europe".

**Data Availability Statement:** Not applicable.

**Acknowledgments:** The authors thank FRUYPER S.A. (Murcia, Spain) for technical support and the material used for the experiment, and also BioRender (Toronto, ON, Canada) for providing some of the pictures used in Figure 1.

**Conflicts of Interest:** The authors declare no conflict of interest.

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
