# Peer review of "Use of Olive Industry By-Products for Value-Added Food Development"

_agronomy, doi:10.3390/agronomy13030718_

Round 1
Reviewer 1 Report
Dear authors, this article is highly relevant for the revaluation of industry waste, generating an impact on the circular economy, however, I consider it relevant to find the following aspects:
2.1.1. Separation of olive industry by-products.
In this section it is advisable to include the volume from which the separation of the products was carried out, and to describe the process specifically, since it is not found in the supplementary material either.
2.1.2. Development of value-added products
The use of the oil fraction for the generation of new products is specifically described and the percentages used are described; however, it is relevant to include the process used to obtain these products.
Fatty acid profile and content of olive oil by-product
In the results and discussions of this section the fatty acids of higher concentration are indicated, however, it would be relevant to indicate those that were identified in lower concentration, this results are related to the extraction process or it is because the olive tree contains small quantities.
Total phenols:
It is presented as results a content of total phenolic compounds, in which it is indicated there was no statistically significant difference, however in the discussion it is indicated phenols such as quercetin and ellagic acid were found in high concentrations at elevated temperature of the process, therefore if an identification of polyphenolic compounds was performed why it is not presented as part of the results in an image?
According to the results of the sensory evaluation, it would be relevant to include discussion if the content of polyphenolic compounds generated an effect on acceptance.
The same section indicates that there was no significant effect in terms of total phenolic content, why not consider other general techniques such as total flavonoids or identifying polyphenolic compounds by spectrophotometric methods?
Reviewer 2 Report
In this study, the authors decribed the recycling usage, practice and evaluation of pickling residue in the development of new food products with detection of oil composition, total phenolic content etc, which provides good reference processing and utilization of olive industrial wastes.
I recommened the authors add some review information or description of health promoting ingredients as well as their content about olive waste reported in the literatures. Besides, in my opinion, the term "high technoligical food" should be avoided.
Reviewer 3 Report
Line 61: Please consider whether the word "R+D+i" should be put in the content and whether it is important. Given that it does not explain it in detail and emphasizes other points in the manuscript
The introduction is too short, please improve it with more background.
Materials and method: Please provide the sources of all materials used
In the table/result, representation of data should be in always in similar manner. Authors should check again whether use the data upto two decimal (Line 107, 252, 279) or upto three decimals’ points (Line 217).
Description under table 3, table 4: Since it was stated that *, **, and *** were the markers that represented the statistical difference of p < 0.05, p < 0.01 and p < 0.001, respectively, but the experimental findings provided in the table was just the statistic different of p < 0.001.
Improve the results and discussion part with up to date references.
It is recommend to Authors add schematic figure to show how circular economic/business works
Authors should mention the limitation of their study. What is the future prospect to overcome the limitation of the research.
Conclusion: I appreciate that your conclusion is so comprehensive. However, it is recommended to summarize more concisely. Some information should be explained in the discussion section instead.
Reviewer 4 Report
Comments and Suggestions for Authors
The paper presents an interesting work on the extraction of high-value-added compounds from residues resulting from the industrialization process of olives. These residues are rich in fatty acids, phenolic compounds, and other substances and they can be considered raw material for the development or enrichment of other foods. The results obtained pointed out that the physicochemical and functional characteristics of the oil and the brine, generated as by-products in the olive industry, turned them into potential raw materials of high technological value. For this reason, new formulations of the filling of olives were developed, using residual oil as a by-product, which showed greater global acceptability by consumers compared to commercial stuffing made with sodium alginate. The paper is both well-structured and written. For this reason, it can be accepted but moderate English revision is required.
Specific comment
• Line 54-56: “Several research have been carried out about the extraction of phenolic compounds through membrane technologies for their application in the food, cosmetic and/or pharmaceutical industry” âž” Please, could the authors add some references about these researches?
• Line 56-58: “but at an industrial level no work is being done on the use of by-products (oil and brine) for incorporation into the same production process or in the development of new products” âž” This is true if we considered only oil and brine. Di Nunzio et al. (2020) (10.1016/j.foodres.2019.108940), for example, studied the effect in a biological system of different bakery products (biscuits and bread) enriched with defatted olive pomace (DOP). The effects of supplementation with the digested bakery products were investigated by measuring cytokines secretion and evaluating the modification of cell metabolome by nuclear magnetic resonance (NMR) spectroscopy.
Reviewer 5 Report
The manuscript does not present microbiological evaluations.
Since this study aims the use of by-products in the development of food products, the microbiological quality of the by-products should also have been evaluated.
Round 2
Reviewer 3 Report
The author responded to my question and I have no further questions